# Use of sediment dwelling bivalves to biomonitor plastic particle pollution in intertidal regions; A review and study

**L. I. Bendell**[1]*, **E. LeCadre**[2], **W. Zhou**[3]

**1** Ecotoxicology Research Group, Department of Biological Sciences. Faculty of Science, Simon Fraser University, Burnaby, Canada, **2** PolyTech Clermont-Ferrand, Aubiere, France, **3** Department of Chemistry, Faculty of Science, Simon Fraser University, Burnaby, Canada

* bendell@sfu.ca

**Data Availability Statement:** All relevant data are within the manuscript and its Supporting Information files.

**Funding:** Funding from the National Research Council of Canada (NSERC Grant # 31-611307 to LB) is gratefully acknowledge. The funders had no

## Abstract

We explore the possibility of using the varnish (*Nutallia obscurata*) and Manila (*Venerupis philippinarum*) clams as biomonitors of microplastics (MPs) pollution. A short review is first provided on the use of bivalves for biomonitoring MPs in aquatic ecosystems. From the conclusions drawn from our review we determine if the sediment dwelling varnish and Manila clam could possibly be good choices for this purpose. We sampled 8 intertidal sites located within two distinct regions of coastal British Columbia, Burrard Inlet (5 sites) and Baynes Sound (3 sites). Each intertidal region had its own particular use; within Burrard Inlet, BMP a heavily used marine park, CP, EB, J, and AP, popular local beaches, and within Baynes Sound, Met and NHB, two intertidal regions heavily exploited by the shellfish industry and RU an intertidal region with limited aquaculture activity. Microfragments were recovered from bivalves collected from all intertidal regions except for AP. Microspheres were recovered primarily from bivalves sampled from Baynes Sound at NHB where high numbers of spheres within sediments had previously been reported. BMP and Met had the highest number of particles present within individual clams which were predominantly high density polyethylene (HDPE) and a polypropylene composite (PPC). Both polymers are extensively used by the shellfish industry in all gear types, as well as in industrial and recreational marine activities. The spatial distribution of recovered MPs was indicative of the anthropogenic use of the intertidal region suggesting these bivalves, for microfragments and microspheres, may be suitable as biomonitors and could prove to be useful tools for determining whether reduction policies for plastics use are having a positive effect on their release into marine environments.

## Introduction

The past 20 years has seen a dramatic increase in our knowledge with respect to microplastics within our marine environments. The science has originated from first coining the term microplastics [1] to documenting their presence in all levels of marine life [2,3] to mapping the extent our marine regions have been impacted by this unique pollutant [3,4].

role in study design, data collection and analysis, decision to publish, or preparation of the manuscript.

**Competing interests:** The authors have declared that no competing interests exist.

Because of the significant negative impacts these plastics are having on our marine ecosystems, governments such as Canada are responding by putting polices in place to reduce and eliminate the flow of plastics from terrestrial to marine systems. Perhaps the most progressive is the recent Oceans Charter [5] led by Canada with the ultimate goal of eliminating and recovering 100% of plastics by 2030.

Given these progressive moves on the management of marine waste, an immediate need is to identify an appropriate biological monitor that would be able to determine if reduction policies are indeed having the desired effect. This is important information for the creation and verification of policy as such a biomonitor would be able to assist in directing the most effective plastics reduction strategies for the protection of our marine ecosystems. Questions that are immediately raised are: What type of plastic to monitor for and what organism would be the most suitable as a biomonitor for the type of plastic that is to be monitored?

## What type of microplastic to monitor for?

An important consideration in the selection of an appropriate biomonitor is that it will indicate trends over time for the microplastic that has been targeted for elimination. For example, the Canadian government has committed to the ban of single use plastics by 2021 [6]. One of the greatest users of single use plastics is the food packaging industry and its associated products such as plastic cutlery, straws and plastic bags. An appropriate biomonitor would then be able to determine if the policy to reduce such plastics use has indeed been effective.

Of the primary shapes of MPs that are recovered from environmental media, fibers, fragments and spheres predominant [3].

## Microfibers

A number of studies have suggested that microfibers are the most abundant form of MPs within environmental media [e.g., 7,8] attributed to multiple sources; finfish and shellfish aquaculture infrastructure (e.g., ropes, netting, cages), plastic tarps (also used by the aquaculture industry), car tires and textiles (e.g., laundry water). Microfibers are a ubiquitous contaminant occurring both in the water column and sediments. And, importantly, unlike microspheres and microfragments, microfibers are an airborne contaminant. This trait makes the analysis for the presence/absence of microfibers within environmental media subject to contamination. Indeed, the recognition that samples can become readily contaminated with microfibers has become one of the most important methodological challenges encountered as we move forward in our understanding of microplastics within our environment. Lachenmeier et al. [9] reported on the extent of the problem and concluded that even with specialized clean rooms, there was still background contamination of samples. Dimitrijevic [10] indicates that even with the strictest of protocols in place, background contamination of microfibers still occurs.

Further, unlike microfragments and spheres, microfibers stay suspended within the water column for extended periods of time. The water column is subject to currents, wave action and storms resulting in a well mixed medium with the transport of microfibers far from their point of origin. Desforges et al. [11] demonstrated that while microfiber concentrations generally decreased as distance from shore increases, microfibers also accumulated within bottlenecks creating "hot spots" of microfibers from multiple sources making it impossible to discern the point of origin of the microfiber. These two factors combined, susceptibility of samples to airborne contamination and their wide dispersal characteristics, would suggest that microfibers would not be the most suitable microplastics to monitor to assess the effectiveness of reduction policies.

## Microfragments and microspheres

By contrast, microfragments and microspheres are found predominantly in sediments and will sediment out close to the point source of origin. Schwarz et al. [12] reviewed the sources, transport, and accumulation of different types of plastic litter in aquatic environments with the goal of obtaining a global framework of plastic waste transport and accumulation needed for plastic pollution mitigation strategies in aquatic environments. These authors concluded that plastic waste transport was affected by particle density, surface area and size, that the predominant polymers were polyethylene and polypropylene and that the larger fraction of plastic litter is likely retained in sediments. Avio et al. [13] also note that the sediment environment is more indicative of microplastics pollution than the seawater column. Erni-Cassola et al. [14] applied a meta-analysis to determine the distribution of plastic polymer types in the marine environment. Their analysis revealed that the most abundant polymer was polyethylene as found by Schwarz et al. [12] followed by PP&A (polyesters, (PEST), polyamide, (PA), and acrylics) and then polypropylene. They also found that all common polymer types were most enriched in intertidal sediments and surface waters concentrations were four orders of magnitude lower than in intertidal sediments. As there is minimal airborne contamination of fragments and spheres and that these MPs settle out within sediments close to their point of origin, these two particle types may be the most indicative of any changes in sources of MPs over time.

## What organism would be best suited as a biomonitor?

The ideal biomonitor is globally distributed, abundant and its response to the desired contaminant of interest indicative of trends of that particular contaminant. For these reasons, perhaps the most popular biomonitor currently in use is the blue mussel (*Mytilus edulis/trossolus*) [15]. For 45 years the blue mussel has been an effective biomonitor of lipophilic organic contaminants (e.g., pesticides), inorganics (e.g., metals such as cadmium and lead) and pharmaceuticals. Hence it has been suggested that this organism could possibly be suitable for the monitoring of changes of microplastics within the water column over time [15]. However, both recent field and lab research has indicated that the use of the blue mussel as well as the Eastern oyster (*Crassostrea virginica*) as a biomonitor of MPs pollution is highly problematic.

## Field research

Brate et al. [16] applied *Mytilus* spp. as sentinels for monitoring microplastic pollution in Norwegian coast waters. Mostly microfibers were found (83%), cellulosic was the most abundant microfiber, not plastic and the authors concluded that uncertainties such as mussel size, the role of depuration and other fate related processes needed further research. Phuong et al. [17] determined microplastic contamination in the blue mussel and Pacific oyster (*Crassostrea gigas*) along the French Atlantic coast. Both fragments and fibers were found and 8 different polymer types of which polypropylene and polyethylene were the most abundant (85%). However, numbers of MPs within each individual were independent of site, season and/or mode of life. Avio et al. [13] assessed the differences in microplastic pollution after the removal of the Costa Concordia wreck using both fish and transplanted mussels. Conclusions included that MPs pollution was more evident in the benthonic environment rather than in the seawater column and that the transplanted mussels had limited capability to discriminate microplastic pollution around the wreck area.

## Laboratory research

Woods et al. [18] studied plastic microfiber uptake, ingestion, and egestion rates in the blue mussel and found that most of the fibers (71%) were quickly egested as pseudofeces with only 9% being ingested. Fernandez and Albentosa [19] studied the uptake, elimination and accumulation of microfragments (irregularly shaped particles of high-density polyethylene) by mussels and found that 83% of particles were cleared from the animal after 6 days of depuration. These authors concluded that their study emphasised the gap of knowledge on the feeding behaviour of mussels in relation to MPs, in this case, fragments and the necessity to investigate it in different marine species and under different exposure regimes. Ward et al. [20] unequivocally showed that the selective ingestion and egestion of plastic particles by the blue mussel and Eastern oyster *(C. virginica)* precludes their use as bioindicators of MPs pollution. These authors advised, given recent proposals to adopt bivalve molluscs that filter feed from the water column as bioindicators of MPs pollution (e.g., [15]), the proposed organisms should ingest without bias the majority of plastic particles to which they are exposed. Their experiments indicated that both the mussel and oyster did show bias in particle selection and thus would be poor indicators of MPs pollution in the environment and recommended that other marine species be explored.

Hence in choosing a biomonitor to assess changes in amounts of plastics debris entering our aquatic ecosystems, selecting a species that does not discriminate the type of material that is being ingested through pseudofeces production, would be important. And, selecting a sediment dwelling species that would be exposed to those MPs accumulating within the surface sediments would be a good first choice. Finally, focusing on microfragments and microspheres would alleviate the analytical challenges that are present for microfibers as there is no airborne contamination and minimal contamination from other external sources (e.g., lab apparel). Sample analysis can occur within a general lab setting without the need for strict contamination protocols. This in turn would allow for sample collection and analysis by most labs so that biomonitoring efforts could be maximized globally.

Both the varnish (*Nuttallia obscurata*) and Manila *(Venerupis philippinarum)* clam are infaunal sediment dwelling bivalves, that both filter and deposit feed exploiting food sources within sediments as well as at the sediment water interface [21,22]. Varnish clams can be found up to 30 cm in depth [21] and the Manila will burrow to a sediment depth that is dependent on its size (small versus large) and sediment characteristics i.e., ease of burrowing and energy required to burrow [23]. Unlike epifaunal bivalves such as oysters and mussels which respond to increased levels of phytoplankton and detritus in the water column with increased filtration capacity and production of pseudofeces and selective feeding, infaunal bivalves such as clams adjust their clearance rates [24]. And unlike oysters and mussels who are highly selective at filtering particles from the water column, varnish and Manila clams feed on a variety of food sources including phytoplankton, benthic diatoms and organic matter from surface sediments, and seagrass and seaweeds. Gillespie et al. [21] note that stomach contents of the varnish clam have been found to contain diatoms, bacteria, macroalgal fragments, small wood fragments and silt. They are non-discriminatory feeders that don't egest excess food through pseudofeces.

On the west coast of British Columbia, they are invasive, originating from Asia, the Manila becoming established in the 1940s with the establishment of the oyster industry and the varnish more recently though ballast waters [21,25]. They display key characteristics required for a biomonitor; 1) they are sessile 2) reside in the sediment 3) are opportunistic feeders (i.e., non-selective), 4) abundant and 5) widely distributed. Finally, they are important food sources for tertiary consumers including humans thus incorporating a human food safety component

into the biomonitor. A complete description of the varnish and Manila clam can be found in [21] and [25] respectively.

Here, we determine if these two bivalves could be effective in indicating spatial differences in microplastics pollution (MPs). To meet this objective, we sampled bivalves from 8 different intertidal regions, each with its own unique use. Microplastics fragments and spheres were extracted from the bivalves and related to the use of the particular intertidal region. Through this approach we demonstrate that these two bivalves may be effective organisms for the use of biomonitoring trends in microplastics pollution, particularly fragments and spheres within our marine intertidal environments.

## Materials and methods

### Study sites

Two distinct coastal regions were selected for sampling, Baynes Sound (49.5362˚ N, 124.8393˚ W)) and Burrard Inlet (49.3174˚ N, 123.1913˚ W) (Fig 1a).

### Baynes Sound

Baynes Sound is located within the biologically significant region of the Salish Sea (Fig 1b). Its physical features include being a thermally stratified inland sea with soft substrate. It is a key location for marine birds and supports globally significant numbers of migratory birds. It

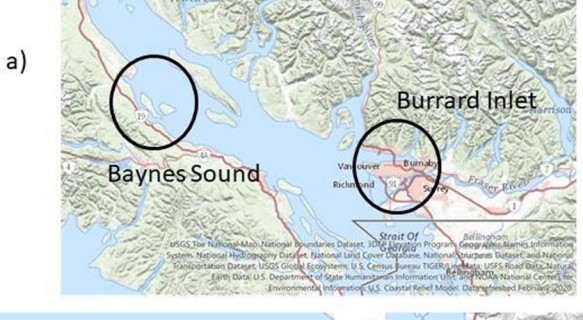

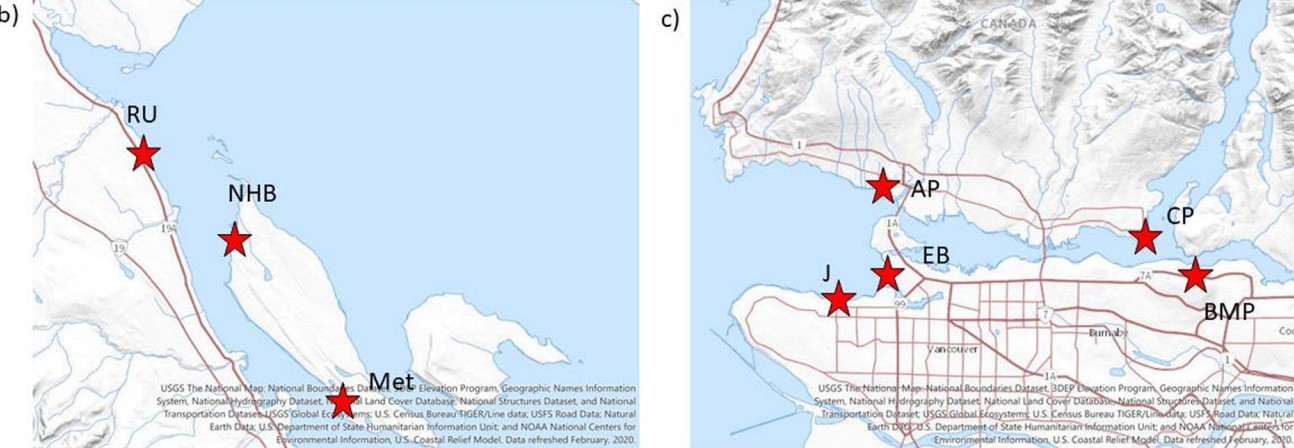

**Fig 1. Locations of the 8 intertidal regions.** Insert a) shows location of the two study areas. b) Baynes Sound, RU, NHB and Met, c) Burrard Inlet, J, EB, AP, CP and BMP. GPS coordinates for each intertidal region provided in Table 1.

serves as a foraging area and haul out for Stellar sea lions and is a key spawning and rearing area for forage fish. These combined features make Baynes Sound one of the most unique and biologically sensitive regions along coastal British Columbia. Because of its unique nature, in 2012, the Canadian Department of Fisheries, Oceans and the Coast Guard (DFO) categorized this region as an Ecologically, Biologically, Sensitive Area (EBSA), which requires a special management approach for its protection [26]. Despite its ecological importance, Baynes Sound is also a region that experiences an expanding and unregulated shellfish aquaculture industry [26]. In 2018 there were 137 tenures located within Baynes Sound. Currently 90% of the intertidal region of the Sound is under shellfish lease with the majority (80%) of the major product, oysters, sold to US markets [27]. Current practices employed by the British Columbia shellfish industry include extensive use of plastics, mostly high-density polyethylene (HDPE) and polypropylene (PP), in all equipment such as trays, rope, netting, buoys, fencing, oyster socks and polyvinylchloride (PVC) piping for geoduck seed [28]. Equipment used by the industry is not secured and quickly becomes derelict fishing gear which washes up onto the shore (Fig 2a–2g). This derelict gear is then subject to ultra violet light and mechanical break-down leading to ever smaller plastic particles that have been shown to accumulate within the sediments of Baynes Sound [29,30]. Since 2004, the local residents of Baynes Sound have held an annual beach cleanup and from 2004 to 2010 approximately 2–3 tonnes of shellfish debris each year was collected. In 2019, twice the amount, 6 tonnes was collected despite the continued efforts of the local community to alert DFO, the Canadian government agency responsible, to the amounts of plastic debris originating from the industry [26,28]. Three intertidal regions within Baynes Sound were sampled for bivalves; Royston/Union, (RU) Metcalf Bay (Met) and North Henry Bay (NHB) (Table 1). Both Met and NHB are extensively used for shellfish farming. Kazmiruk et al. [29] have reported extremely high numbers of microspheres in NHB as well as the presence of fibers and fragments within the sediment of the Sound generally.

## Burrard Inlet

Burrard Inlet is a shallow coastal fjord that runs through the heart of the Greater Vancouver Regional District (Fig 1c) (see [31] for a full account of the Inlet). It is Canada's largest ice-free deep-water port with its industrial history beginning with the logging of the great old growth ca. 1860. The past 160 years has seen the Inlet used for shipbuilding, wood creosoting, manufacturing, and boating of all types (e.g., industrial, pleasure craft and cruise ships) as well as rapid residential development.

English Bay (EB), Jericho (J) and Ambleside (AP) are in the Outer Harbour of Burrard Inlet (Fig 1c). There has been limited industrial development of the outer harbour with major modifications such as seawalls constructed for residential development [31]. EB and J are both sandy beaches used primarily for swimming. EB however is located at the base of one of the highest density urban regions of Vancouver with a major traffic route just adjacent to the beach. It is also the location of the "Festival of Light" an international fireworks competition that occurs each year in July. AP is a rocky cobble beach with a seawall walkway that limits public access. Both Cates Park (CP) and Barnett Marine Park (BMP) are located within the Central Harbour of Burrard Inlet and are both sandy to cobble beaches (Fig 1c). CP is located on the north side of the inlet and has minimal direct impacts. By contrast, BMP is located directly adjacent to industrial (oil refinery) and port use. It also receives effluent from an authorized industrial outfall, plus a combined sewer outfall, a sanitary sewer outfall and several stormwater outfalls. Of the 5 intertidal regions sampled within Burrard Inlet, BMP has been the most heavily impacted by anthropogenic activities [31]. Indeed, the most severely impacted

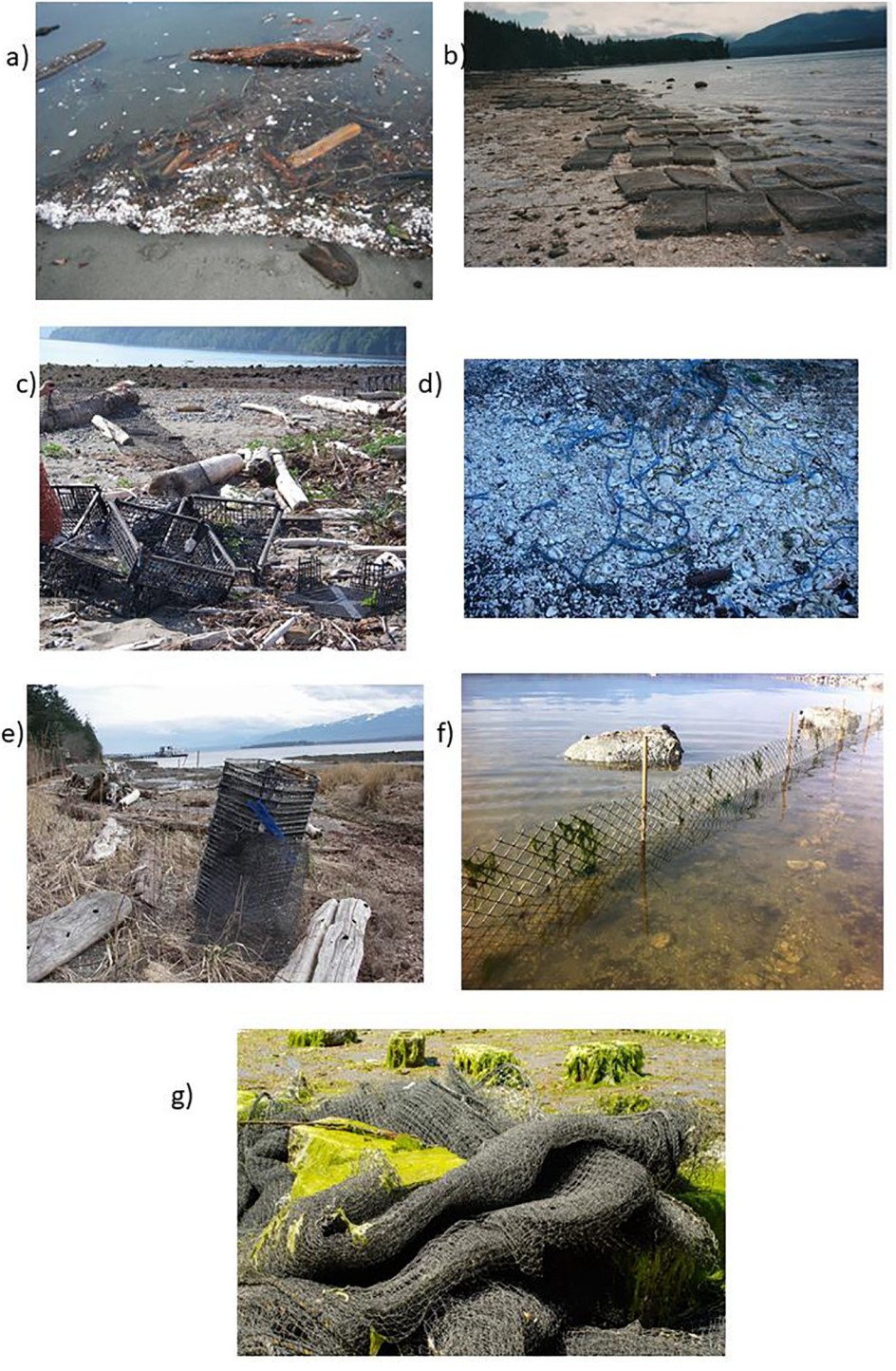

**Fig 2. Examples of derelict shellfish gear recovered from the intertidal regions of Baynes Sound.** Examples are of a) polystyrene, b) oyster pouches, c) oyster baskets and mesh, d) rope, e) oyster trays and mesh, f) oyster fencing and g) anti-predator netting. Note the broken basket in c). http://adims.ca/photo-gallery/.

**Table 1. Characteristics and GPS coordinates for the 8 intertidal regions.**

| Intertidal region | Use | Substrate type | GPS |
|---|---|---|---|
| | | | N˚W |
| BMP | Industrial/recreational | sandy to cobble | 49.1725:122.5530 |
| CP | Recreational | sandy to cobble | 49.3030:122.9553 |
| J | Recreational/residential | sandy | 49.2722:123.1985 |
| EB | Urban/residential/recreational | sandy | 49.2872:123.1614 |
| AP | Recreational | cobble | 49.3234:123.1500 |
| Met | Industry | cobble | 49.4876:124.7506 |
| NHB | Industry | sandy to cobble | 49.5817:124.8331 |
| RU | Recreational | cobble | 49.6083:124.8990 |

region of Burrard Inlet is the Port Moody Arm, which lies just east of BMP. BMP is also located just below the Burnaby Terminal which is the terminus of the proposed Trans Mountain Pipeline System, a distribution point for crude oil and refined products (https://www.transmountain.com/burnaby-terminal-and-tunnel; accessed November 13th 2019). Order of decreasing anthropogenic impacts for the 8 intertidal regions is, BMP, Met, NHB, EB, RU, J, AP and CP (Table 1).

## Bivalve sampling

During May-June 2019, at each site, between 10–30 bivalves were collected by shovel at 5–10 cm depth within the sediment, (with permission from the Department of Fisheries, Oceans and the Coast Guard Permit # XMCFR 24 2019), placed on ice, transported back to the lab, euthanized by freezing and kept frozen until analysis. Both varnish and Manila clams were recovered from J, varnish from BMP, CP and EB and Manilas from AP, Met, RU, and NHB.

## Plastics extraction

Prior to extraction, clams were partially thawed and length, height and width recorded. Tissue was carefully extracted and wet weight obtained for each bivalve. No pooling of samples was done as we wanted to determine the number of MPs within each individual bivalve.

There have been a number of procedures recommended for tissue digestion and after a review of the different methods we adopted those suggested by [32]. A 10% KOH at 4 times tissue volume was added to each bivalve within a 50 mL Erlenmeyer flask. Samples were incubated at 60˚ C for 48 hours. After 24 hours of incubation 2.5 mL of Rit Dye More® which dyes synthetics was added to help in the identification of plastic materials. After incubation, each sample was filtered through a Whatman™ 1001–090 Grade 1, Pore Size, 11µm qualitative cellulose filter paper. To remove any remaining fatty tissues, filters were rinsed with 30% hydrogen peroxide. Filters were placed into petri dishes for later microscopic examination. All procedures were carried out within a laminar flow fume hood. Each filter was analysed at 12-50X magnification for the presence of microplastics. Suspected MPs were recovered, placed onto a slide with double sided tape to retain the particle on the slide and submitted for FTIR analysis for plastics confirmation. Infrared spectra between 550 cm$^{-1}$ to 4000 cm$^{-1}$ were measured on a PerkinElmer FT-IR Microscope Spotlight 200i, focusing on a 10 µm area of the sample with a diamond crystal.

## Quality assurance quality control

Both positive and negative controls were run to correct for potential contamination of our samples by spheres and fragments. As noted in the introduction, due to the significant problem of airborne contamination of samples by microfibers, we did not include these in our analysis. Exceptions were made for those fibers that were obviously not of textile origin (see comments below). Positive controls were run using a face cleanser which contained polyethylene microbeads (Nivea® Face Scrub). The face cleanser was subject to all sample digestion procedures to provide a positive check that methods were effective in the recovery of microplastic spheres. Negative controls, i.e., procedural blanks were run in parallel with each batch of samples, and as with the positive controls, all steps used in the digestion procedure applied to the negative controls. All equipment that came into contact with the samples was rinsed thoroughly with filtered de-ionised water prior to use. Latex gloves and cotton laboratory coats were worn throughout. We were successful in controlling for both microsphere and microfragment contamination with none recovered in our blanks. As expected, microfibers were problematic. As our lab was not set up as a specialized clean room, required for the analysis of microfibers in environmental media we focussed solely on microfragments and spheres and only included microfibers that were that were large (ca. 0.5 mm length and would not be air-borne) and could be visually assessed that they were not of textile origin.

## Statistical analysis

As data was normally distributed (Normality Test; Shapiro-Wilk; P > 0.05) no transformations were required. A one-way ANOVA was applied to determine differences in weight, length, and height of bivalves and the number of particles per bivalve collected from the 8 intertidal regions. Sites within each region were then pooled and a simple student's t-test applied to determine if the number of plastic particles recovered from the bivalves differed between the two regions. Acceptance of significance was P<0.05.

# Results and discussion

## FTIR analysis; assigning polymer type

The application of FTIR analysis to identify the nature of the plastic polymer has become a required procedure in assessing MPs pollution. Typically, suspect plastic samples are collected from the matrix and submitted to an external lab for FTIR analysis with assigning of the polymer type arrived at by a comparison of the sample spectra to a library of reference spectra either available as software or accessible online (e.g., Bio Rad® Knowitall®, SBDS). FTIR spectra are assigned based on a "best match" and accompanied with a score based from 0–100 that provides a measure of "confidence" in the assignment of the sample. In most cases a value of 80% is accepted as a good level of confidence.

Complicating the use of spectral data bases for polymer identification relates to the weathering of the polymer. Exposure to ultra violet light, mechanical abrasion, biofouling, ingestion and digestion by the animal, and finally, sample digestion within the lab (10% KOH and 30% hydrogen peroxide) will all serve to degrade the polymer from its original state and thereby its original spectra used in spectral data bases for polymer identification.

Two examples of polymer weathering are given in [33] and [34]. PP composites (PPC) were exposed to natural weathering (rain, sunlight and wind), and the polymer sampled every 2 months for FTIR analysis for 6 months [33]. Through FTIR analysis it was observed that weathering, i.e., photooxidation, caused the oxidation of the PPC resulting in the presence of carboxylic acid and ketone species altering the original PPC spectra [33]. Rajakumar et al., [34]

also studied the natural weathering of PPC within the context of climatic conditions. Samples of PPC were exposed to the natural environment during the summer (80 days) and winter (50 days) months. Samples were taken every 5 days for FTIR analysis to observe the evolution of oxidation products over time. These authors also found the formation of products such as ketones and esters which altered the spectra from its original form. Notable were increased absorbance's at 1700 cm$^{-1}$ and 3300 cm$^{-1}$ which would create significant confusion in the assigning the polymer to its actual polymer type [34]. Comparing weathered particles to a reference data base could lead to misidentification and possibly the under reporting of the number of plastic polymers actually recovered from the environmental media.

To avoid the potential for misidentification and possible under reporting, we applied the following approach; through a review of the recent literature we identified the top most common plastic polymers that have been recovered from environmental media [3,12,35]. In general the order of abundance from most to least, include polyethylene (High Density PE, Low Density PE), polypropylene (PP), polystyrene (PS), polyamide (nylon), polyester (PES), polyvinylchloride (PVC), polyethylene terephthalate (PET), acrylonitrile butadiene styrene (ABS, rubber), ethylene-vinyl acetate (EVA), polymethylmethacrylate (PMMA), and polyurethane (PU). A visual reference library of these polymers was created from a number of sources (e.g., [33], Bio Rad®, Knowitall® [35]) and characteristic sorption peaks for each polymer were tabulated into a matrix (Table 2). Examples of spectra used in the FTIR assignment are presented in S1 Fig. As we were also determining if derelict shellfish aquaculture gear was a source of MPs to the bivalves (one of the anthropogenic activities that we were monitoring for (e.g., Fig 2a–2g), FTIR spectra of the most common plastics used by the industry were determined and

**Table 2. Matrix of characteristic sorption peaks of the main polymers of interest.** Sources include Jung et al. [35], Bio-Rad®, Knowitall® and SDBS.

| Polymer | Wavelength cm-1 | | | | | | | | | | | | | | | | |
|---|---|---|---|---|---|---|---|---|---|---|---|---|---|---|---|---|---|
| HDPE | | 2915 | | 2845 | | | | 1472 | | | | | 730 | 717 | | | |
| LDPE | | 2900 | | 2845 | | | | 1456 | 1377 | | | | 730 | 717 | | | |
| | | | | | | | | | | | | | | | | | |
| PP | | 2950 | 2918 | 2836 | | | | 1456 | 1376 | | 1000 | 842 | 800 | | | | |
| PS | 3446–3026 | | | 2846 | | | 1608 | 1492 | | | 1027 | | 700 | | | | |
| polyamide | 3285–3066 | 2932 | | 2860 | | | 1634 | | | | | | | | 680 | 570 | |
| PES | 3423 | 2969 | 2917 | | | 1711 | | | | | 1093 | | | 718 | | | |
| PVC | | 2911 | | | | | | 1430 | 1331 | 1250 | 1000 | | | 718 | | | |
| PETE | | | | | | 1730 | | | | 1240 | 1096 | 872 | | 720 | | | |
| ABS | | 2922 | | | | | 1602 | 1490/52 | | | 1000 | | 750 | | | | |
| EVA | | 2917 | | 2848 | | 1740 | | 1469 | | 1241 | 1020 | | | 720 | | | |
| PMMA | | 2992 | 2949 | | | 1721 | | 1433 | 1386 | 1238 | 1189 | 985 | 750 | | 638 | 554 | 509 |
| PU | | | | 2856 | | 1731 | 1531 | 1451 | | 1223 | | | | | | | |
| | | | | | | | | | | | | | | | | | |
| cellulose | | | | | | 1735 | | | 1366 | | 1030 | | | | 600 | | |
| | | | | | | | | | | | | | | | | | |
| | | | | | | | | | | | | | | | | | |
| oyster tray | | 2921 | | 2847 | | | | 1476 | | | | | 729 | | | | 414 |
| fencing | 3334 | 2916 | | 2848 | 2364 | | 1656 | 1474 | | | 1046 | | | 719 | | | |
| rope | 3332 | 2952 | 2920 | | 2358 | | | 1462 | 1383 | | 989 | 838 | | | | 480 | |
| pouch | 3331 | 2916 | | 2847 | 2349 | | 1647 | | | | 1076 | | 725 | | | | 470 |
| netting | 3332 | 2925 | | 2794 | | | 1650 | 1462 | | | 1050 | | | | | | 450 |
| tube | | 2925 | | 2831 | | | 1631 | 1481 | | | 1069 | | 731 | | | 487 | |
| basket | | 2921 | | 2847 | | | | 1476 | | | | | 729 | | | | 450 |
| mesh | 3338 | 2943 | | 2831 | | | 1650 | 1462 | | | | | | 700 | | | |

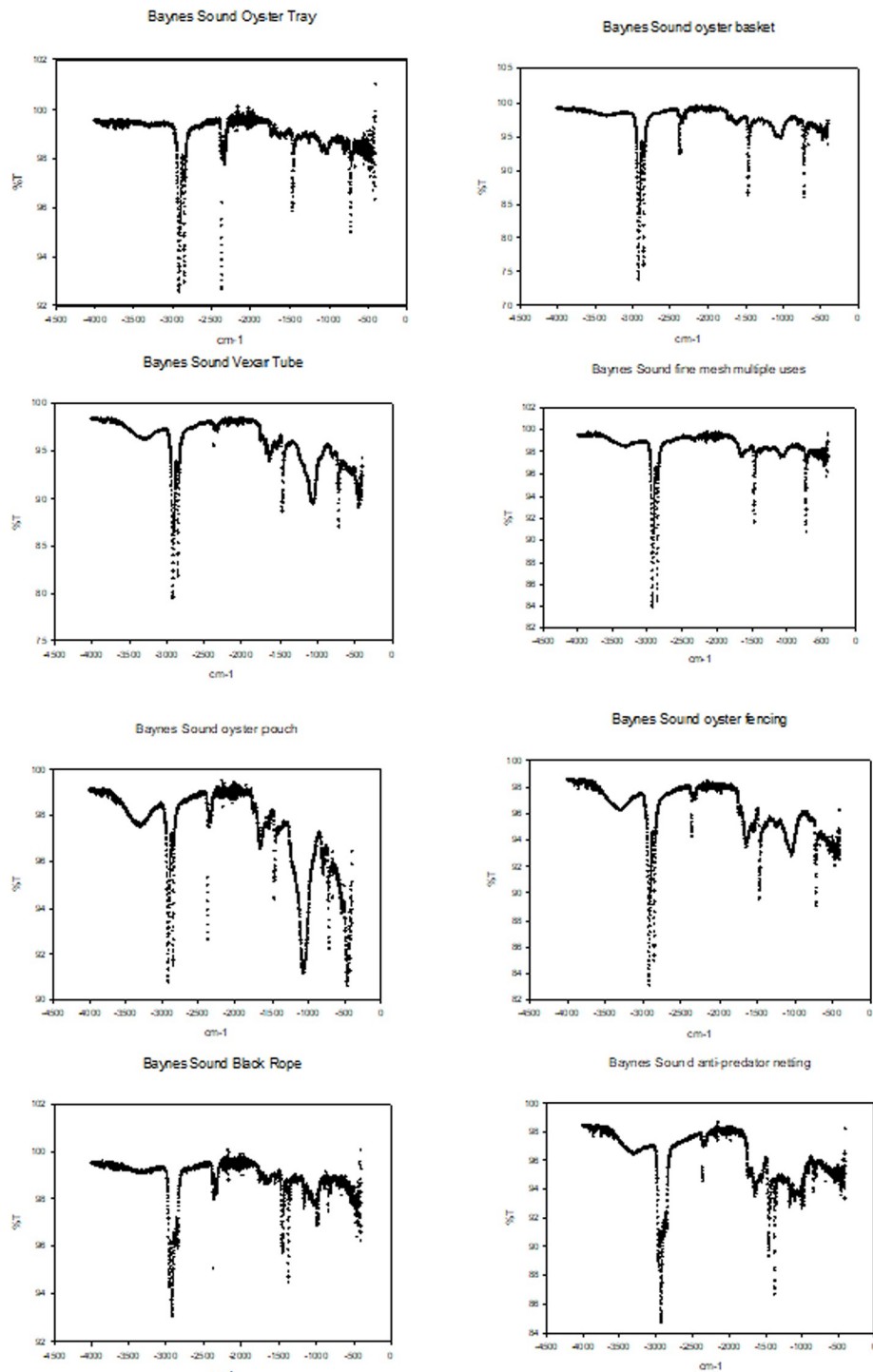

**Fig 3. Reference spectra for shellfish aquaculture gear collected from the intertidal regions of Baynes Sound.** a) oyster tray, b) oyster basket, c) Vexar tube, d) mesh, e) oyster pouch, f) oyster fencing and g) anti-predator netting. Oyster tray, basket and mesh are HDPE and the Vexar tube, pouch and fencing, rope and anti-predator netting are PPC.

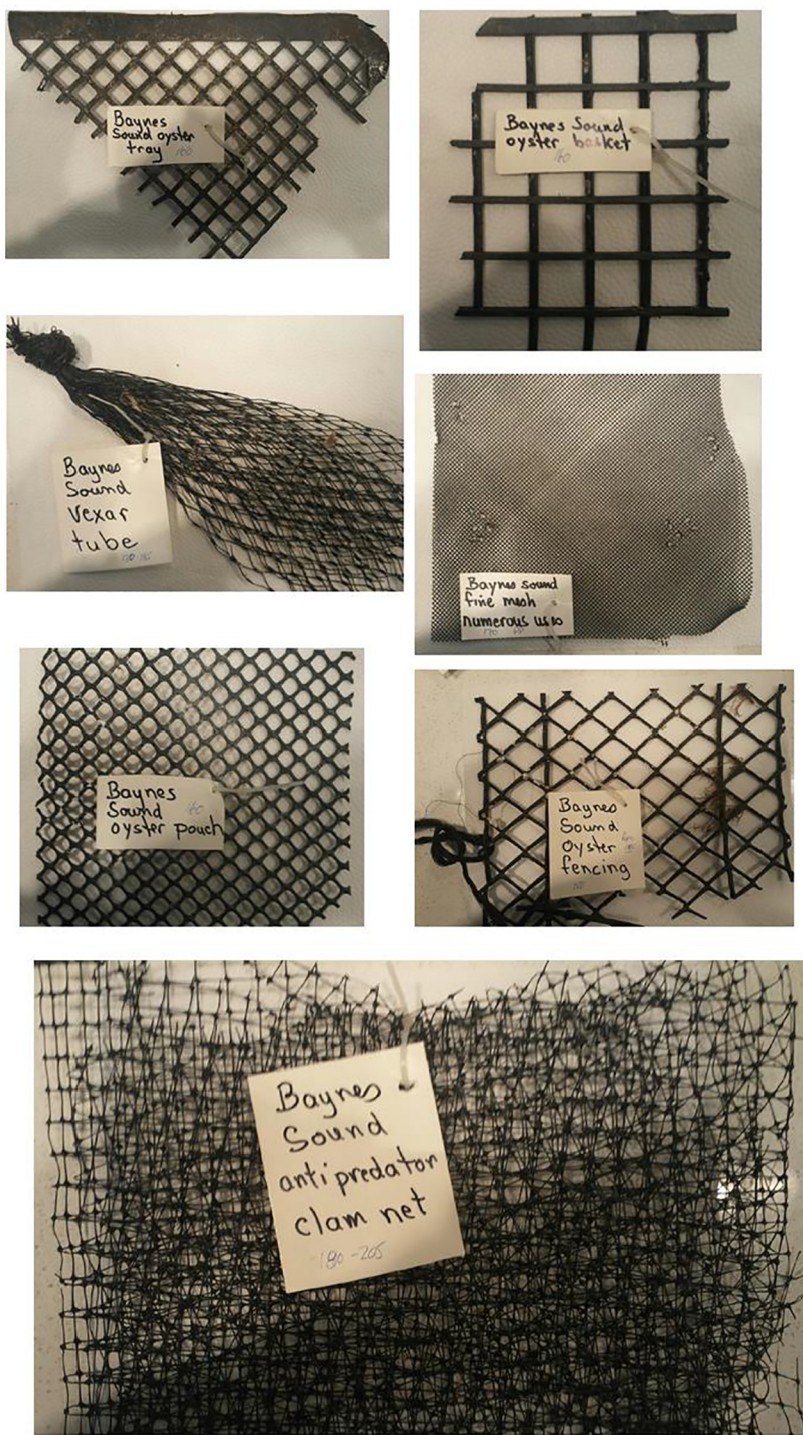

**Fig 4. Shellfish aquaculture equipment collected from the intertidal regions of Baynes Sound.** Matching spectra are presented in Fig 3a–3g.

also included in the matrix of characteristic sorption peaks (Table 2, Figs 3a–3g and 4a–4g). Also included were spectra for those particles most commonly misidentified as plastic e.g., sand, calcium carbonate and cellulose (Table 2, S1 Fig).

Spectra obtained from each of our samples (165 submitted), were individually plotted, %T versus cm$^{-1}$ (SigmaPlot$^{®}$) with "like" spectra placed into groups. Nine distinct spectra were identified. The distinct spectrum from each group was then matched visually (S1 Fig) as well as by comparing characteristic sorption spectra to those compiled in Table 2. Hence rather than rely on software which accesses 1000s of spectra and subject to misinterpretation due to weathering of the polymer, we targeted the most abundant plastics polymers that have been recovered in environmental media and determined if the spectra of particles recovered from the bivalves matched those spectra of these most abundant polymers.

Through this approach 6 polymers were identified, PE, PP, Nylon, polystyrene (PS), polyurethane (PU) and polymethylmethacrylate (PMMA). Also identified was Cyanox 53. Spectra also matched to sand and CaCO$_3$. Thirteen spectra could not be assigned. Microfragments were the primary shape recovered and the most abundant polymer type was HDPE (42/140; 43%) followed by PP (22/140; 22%) (Fig 5a and 5b). The other 4 polymers, Nylon, PMMA, PS, PU, and Cyanox 53 occurred in similar numbers (ca. 4/140; 3%). Sand and CaCO$_3$ were identified in 9 and 3 samples respectively.

## Use of the varnish and Manila clam as biomonitors of MPs

Clams were present at all sites. Varnish clams from CP and BMH were significantly larger than those collected from the 6 other intertidal regions (Table 3). MPs were present in all clams except those from AP (Fig 5b). Clams from Met contained significantly greater numbers per clam as compared to the other 7 sites followed by BMP (Table 4).

Both these intertidal regions are those that experience the greatest anthropogenic impacts, Met from the shellfish aquaculture industry and BMP from multiple industrial uses. We were also able to match directly the spectra from shellfish aquaculture gear (PE, PP composite) to fragments recovered from clams sampled from one of our sites (Met) within Baynes Sound (Fig 6a and 6b). Overall number of particles recovered from clams sampled from the two regions were not significantly different (P>0.05, Table 5).

Microparticles were 50–600 μm in diameter and ranged in shape and composition e.g., fibrous, pellet and fragment (S2 Fig and S1 Table). This diversity in size and shape of particle is important as it indicates that the two clams are non-discriminatory in their feeding. A most interesting and unique finding was the presence of Cyanox 53 an antioxidant, in varnish clams collected from J and EB (Fig 7a and 7b).

LeCadre [36] reports the presence of Cyanox 53 in 11/12 and 7/13 clams sampled from EB and J. Cyanox 53 was also detected in clams from CP (4/10) and AP (3/7). This compound was not detected in clams from Baynes Sound suggesting a unique source to Burrard Inlet. Antioxidants such as Cyanox 53 are used in polyethylene, polypropylene, polystyrene and ABS to prevent oxidation of the polymer which can result in loss of strength, breakdown or discoloration [37]. Although the compound is associated with plastic polymers, it is unknown why it would present as fibrous particles in the clams and indeed the source of this compound warrants further study.

Microfragments recovered from clams sampled from Met could be matched directly to polymers HDPE and a PP composite used by the aquaculture shellfish industry (Fig 6a and 6b). Microspheres were recovered from clams sampled from NHB where [29] reported the presence of extremely high sediment concentrations of microspheres. The authors speculated that given depositional patterns within the Sound, that the high numbers could be a result of

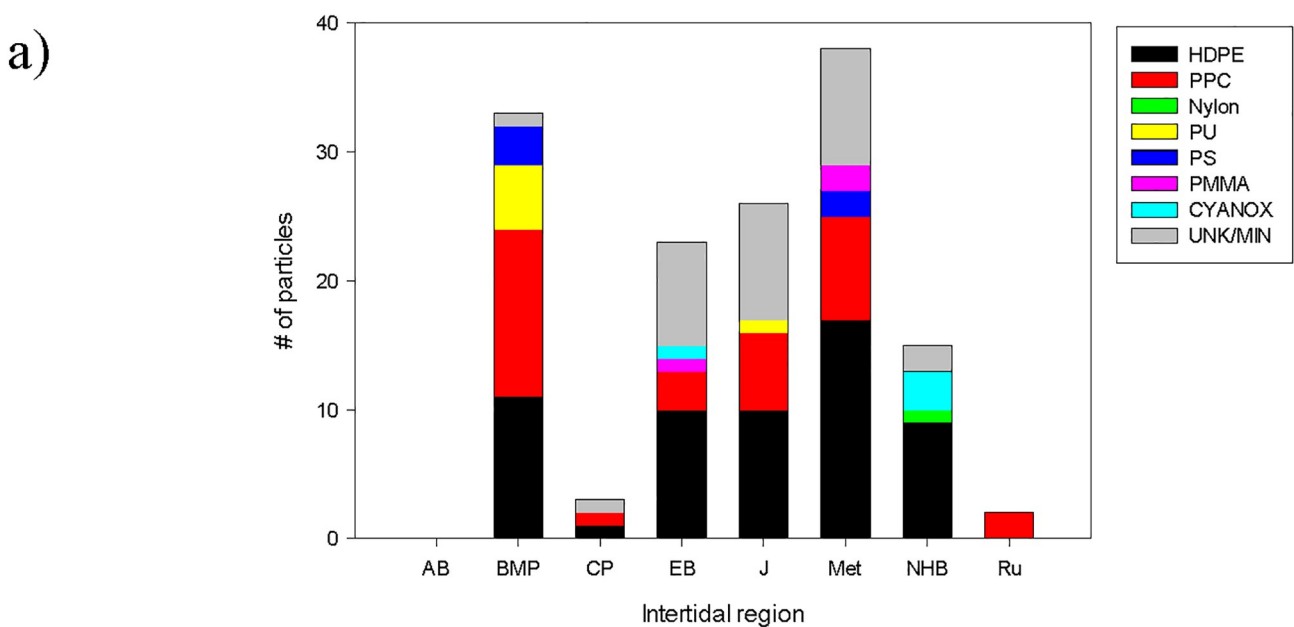

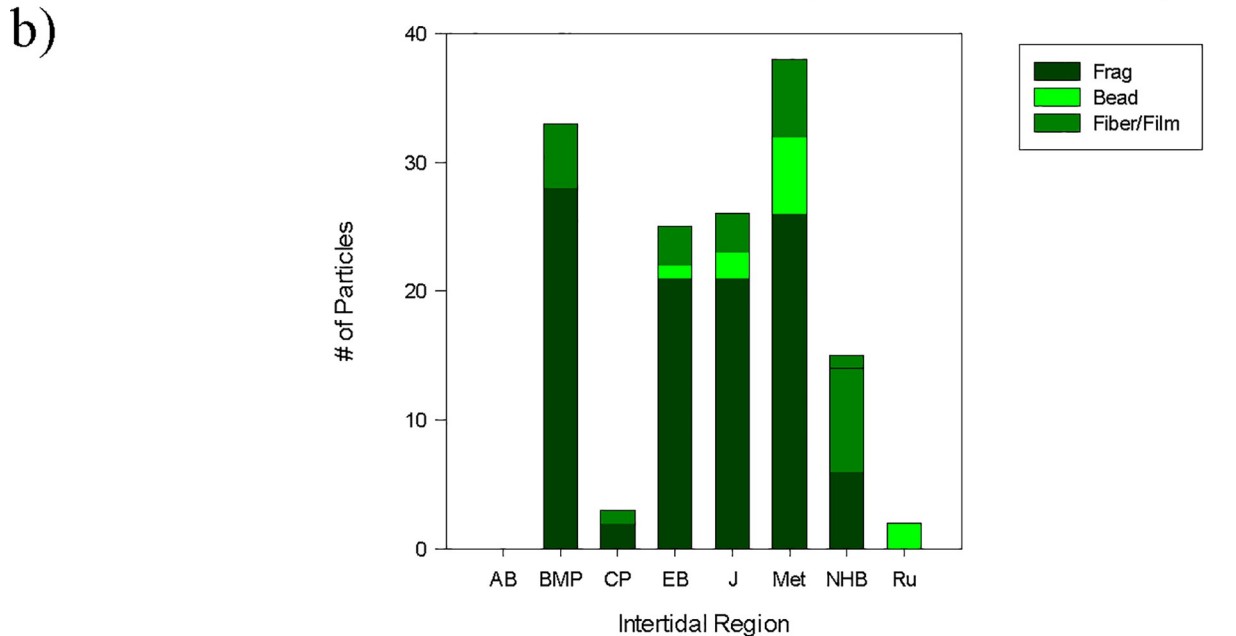

**Fig 5. Polymer composition and shape recovered from the 8 intertidal regions.**

inputs from the Courtney-Comox Estuary which receives effluent from a number of munici-palities, assisted living facilities as well as historically a hospital. FTIR analysis identified the spheres as PE and PMMA. PE spheres are widely used in cosmetics (e.g., Nivea Face Scrub®), which have been banned by the Canadian Government [38]. However, microspheres are

**Table 3. Metrics of clams sampled from the 8 intertidal sites.**

| | Site | | Wet Weight (g) | | Width (mm) | | Length (mm) | | Height (mm) | |
|---|---|---|---|---|---|---|---|---|---|---|
| | | | Mean | SE | Mean | SE | Mean | SE | Mean | SE |
| | AP | 7 | 8.16 | 1.32 | 47.24 | 2.44 | 37.67 | 2.3 | 17.75 | 1.08 |
| Burrard | BMP | **15** | **24.99** | **1.53** | **65.22** | **1.43** | **53.2** | **1.16** | **25.86** | **0.69** |
| Inlet | CP | **10** | **16.49** | **1.06** | **58.27** | **1.66** | **49.11** | **1.22** | **23.03** | **0.56** |
| | EB | 10 | 7.5 | 0.21 | 46.58 | 0.97 | 35.08 | 0.54 | 15.5 | 0.33 |
| | J | 18 | 8.46 | 0.92 | 44.99 | 1.81 | 35.7 | 1.74 | 18.29 | 0.54 |
| Baynes | Met | 13 | 6.77 | 0.48 | 40.85 | 0.81 | 30.41 | 0.64 | 23.19 | 0.82 |
| Sound | NHB | 15 | 6.95 | 0.48 | 40.81 | 0.79 | 30.55 | 0.54 | 22.63 | 0.5 |
| | RU | 12 | 6.91 | 0.39 | 41.47 | 0.58 | 30.47 | 0.65 | 23.09 | 0.55 |
| | | | | $P < 0.001$ | | $P < 0.001$ | | $P < 0.001$ | | $P < 0.001$ |

extensively used by the medical industry e.g., prolonged or controlled drug delivery and to target specific sites at a predetermined rate [39]. They can be made of polymeric waxy or natural and synthetic polymers with sizes ranging from 1–1000 μm. Of the non-biodegradable polymers is polymethylmethacrylate (PMMA) as identified in this study and the source of these microspheres to the environment warrants much further study.

Our findings directly contradict a recent study of Covernton et al. [40] and by doing so, provides a good example of how the inappropriate choice of a biomonitor as made by Covernton et al. [40] can lead to erroneous conclusions. In a study partially funded by the British Columbia Shellfish Growers Association, Covernton et al. [40] attempted to determine if shellfish aquaculture infrastructure contributed to MPs concentrations in bivalves. To do so, they compared MPs concentrations in Manila clams and Pacific oysters (*Crassostrea gigas*) transplanted onto commercial shellfish intertidal leases versus nearby non aquaculture intertidal sites. Bivalves used in the study of Covernton et al. [40] were transplanted from one source, a shellfish farm located within Baynes Sound. Oysters were placed on the surface sediments and Manila clams covered to a 2.5 cm depth within the surface sediment. Oysters and clams were deployed in June and recovered 3 months later and the presence of MPs determined. Three aquaculture sites on the western side of Baynes Sound were included in the study with reference sites located directly adjacent to the aquaculture sites. Covernton et al. [40] reported that recovered MPs from the shellfish were predominately microfibers and found no difference in

**Table 4. Differences in number of particles recovered from clams among the 8 intertidal sites.**

| Region | Site | N | # Particles/clam | | % of clams with particles present |
|---|---|---|---|---|---|
| | | | Mean | SE | |
| | AP | 7 | 0.00 | 0.00 | 0.00 |
| Burrard | BMP | 15 | 2.07 | 0.85 | 53.00 |
| Inlet | CP | 10 | 0.30 | 0.30 | 10.00 |
| | EB | 10 | 2.60 | 0.80 | 80.00 |
| | J | 18 | 1.06 | 0.48 | 28.00 |
| Baynes | Met | 13 | **2.92** | **1.00** | **46.00** |
| Sound | NHB | 15 | 1.00 | 0.43 | 33.00 |
| | RU | 12 | 0.17 | 0.17 | 8.00 |
| | | | | $P = 0.031$ | |

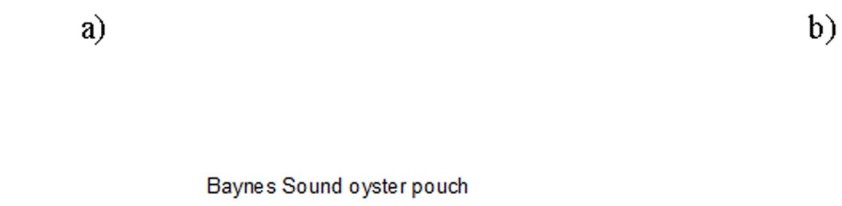

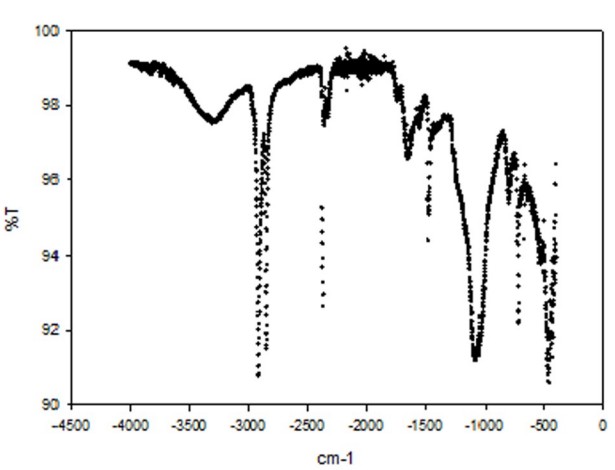
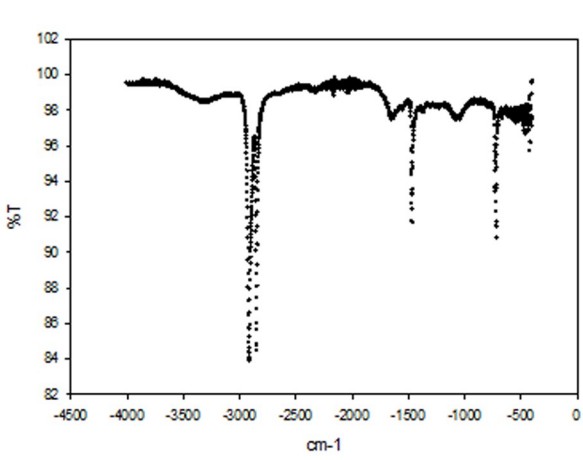

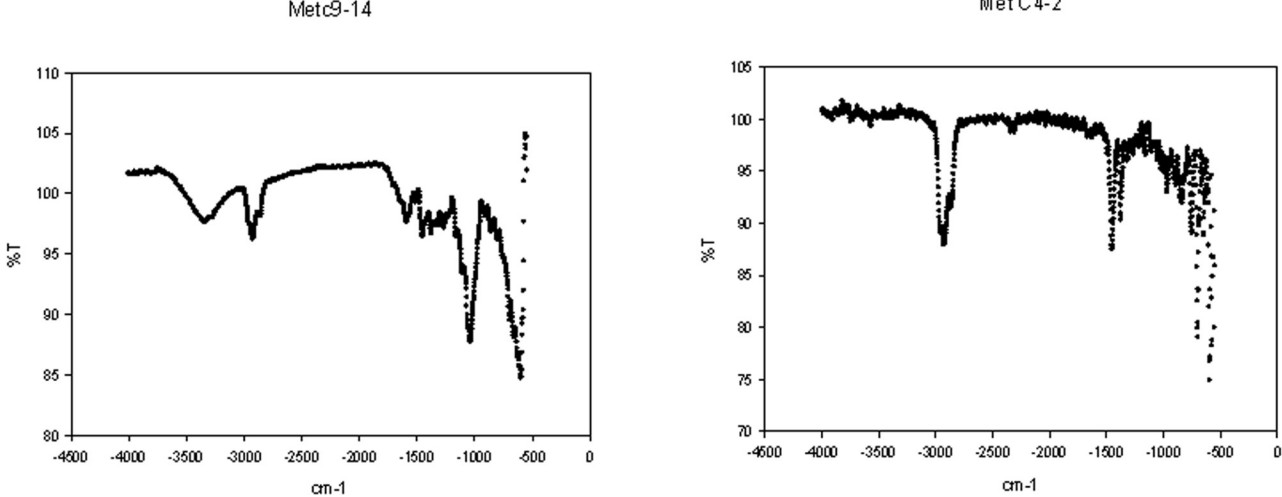

**Fig 6. Polymer identification for particles recovered from clams sampled from Met within Baynes Sound.** Both PP composite a) and HDPE b) are identified.

microfiber concentrations in oysters or clams transplanted onto reference versus aquaculture sites. From this observation the authors concluded that the primary microplastic present within Baynes Sound were microfibers and as determined by FTIR, were of textile origin and not attributed to the degradation of aquaculture infrastructure.

**Table 5. Differences in number of particles recovered between regions.**

| Average number of particles recovered/clam | | | | Average number of particles recovered/region | | | |
|---|---|---|---|---|---|---|---|
| Region | N | Mean | SE | Region | N | Mean | SE |
| Burrard Inlet | 5 | 1.2 | 0.5 | Burrard Inlet | 5 | 17.2 | 6.59 |
| Baynes Sound | 3 | 1.4 | 0.8 | Baynes Sound | 3 | 19.66 | 11.78 |
| | | | P>0.05 | | | | P>0.05 |

As noted within the introduction, the well mixed nature of the water column due to currents, wind and storm events would preclude assigning a particular source of plastic microfiber contamination to the shellfish. More importantly though are the conclusions and subsequent advice of Ward et al. [20] who state that filter feeders such as the oyster, because of their bias in food ingestion cannot be used as biomonitors of MPs thus rendering the observations and conclusions of Covernton et al. [40] with respect to the oyster unsubstantiated.

Covernton et al. [40] also included the Manila clam as a biomonitor of MPs and as with the oysters, found no difference in plastic particles, specifically microfibers, between aquaculture and reference sites. Of the deployed clams (60 at each of 21 sites), insufficient survival for clam recovery occurred at 3 of the reference sites including one in Baynes Sound. No data on the number of clams that survived at each site was provided, nor was a condition index that would have provided information on the health of the clam, i.e., once deployed, was it actually feeding within the sediment. Recovery of a maximum of 10 clams per site retrieved for MPs analysis suggests a high mortality rate although the actual mortality rate at each site was not reported.

Within Baynes Sound, Manilas are found at a depth of 5 cm, necessary to avoid exposure to extreme temperature and rain events. The shallow depth at which the clams were buried may have contributed to the poor survival of the transplanted clams at one of the reference sites within Baynes Sound and the overall low recovery of the Manila clams from the various study sites suggesting that the transplanted clams were not healthy and possibly not actively feeding. Without a condition index it is impossible to know. As sample processing was not done within a clean room required to ensure control of airborne contamination from microfibers, an alternate interpretation of the findings of Covernton et al. [40] is that the textile microfibers recovered from the transplanted shellfish are indicative of the degree of airborne contamination of microfibers within the laboratory where the samples were processed.

By contrast, both varnish and Manila clams sampled in the current study were healthy individuals collected at 5–10 cm depth. They would have been actively feeding up until the time of sampling and MPs recovered from the clams would then reflect what had been ingested up until that time. Unknown are particle retention times within bivalves such as the Manila and whether retention times are shape dependent. For example, for the blue mussel, retention times of 6 days for microfragments and 24 hours for microfibers have been reported [10, 19] suggesting that the behaviour of microfibers within bivalves are much more dynamic than fragments and possibly with a shorter resident time than fragments.

## Conclusions

Both the varnish and Manila clam maybe suitable as biomonitors for tracking trends in MPs concentrations in environmental media that may occur as reduction policies for plastics use are put in place. By focusing on fragments and spheres within surficial sediments and the most abundant polymers (e.g., polyethylene and polypropylene) that are found in marine ecosystems monitoring efforts can be simplified and be undertaken by laboratories globally. Further

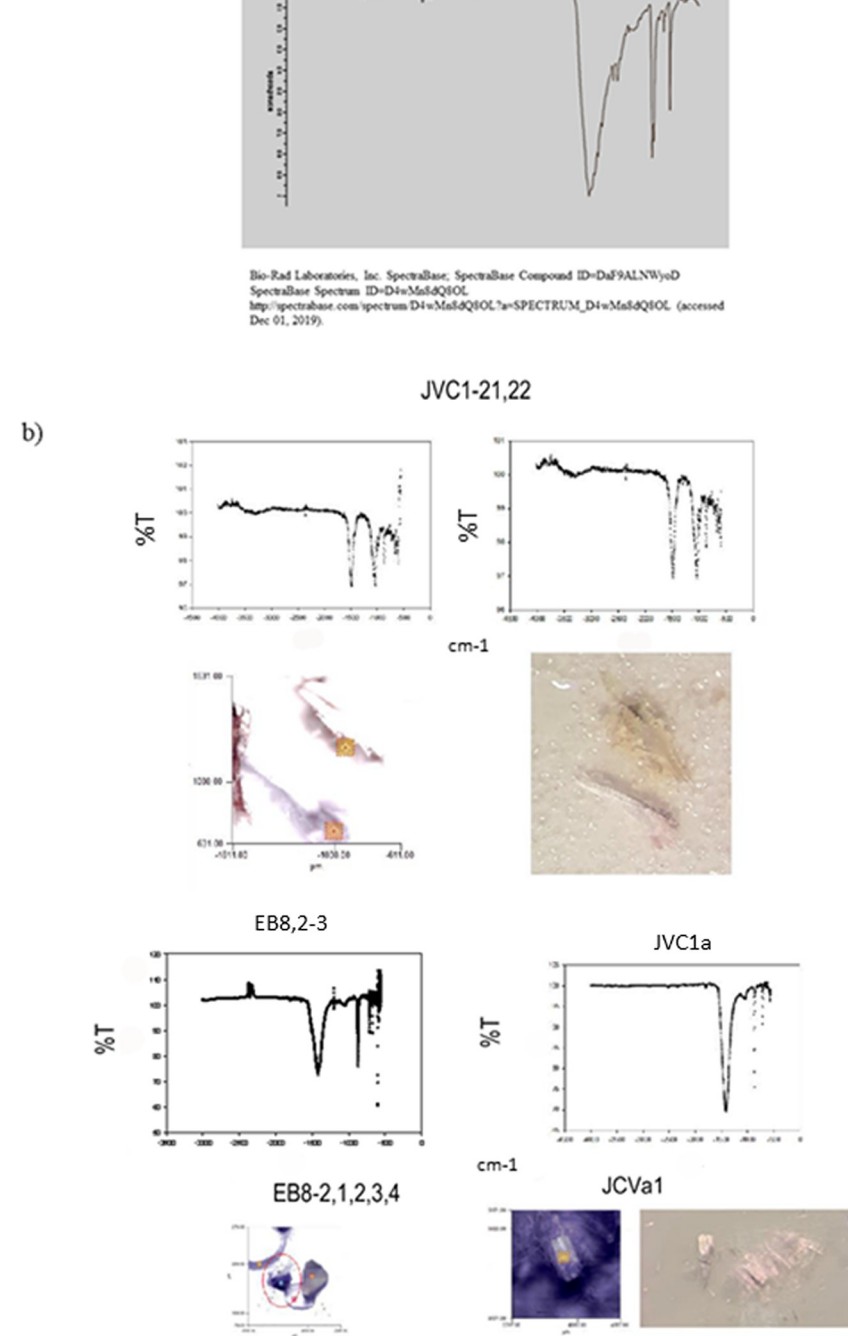

**Fig 7. Cyanox 53 in clams sampled from J and EB.** a) Spectrum of Cyanox 53. b) Spectra of three samples from J and one from EB and images of the fragments recovered.

study though is required to determine basic feeding behaviour of these two species such as particle retention time within the gut and whether retention time is particle type specific. Further understanding of other aspects of the basic ecology such as seasonal effects, food abundance and age would also help assess if these clams will indeed serve as effective biomonitors of MPs.

## Supporting information

**S1 Fig. Reference spectra for polymers identified in the study.** Supplementary material.
(PDF)

**S2 Fig. Particles shape and size recovered from clams sampled from the 8 intertidal regions.**
(PDF)

**S1 Table. Polymer identification for all recovered particles.**
(PDF)

## Acknowledgments

The authors are grateful to C. Young and S. McKeachie for the collection of clams from Baynes Sound.

## Author Contributions

**Data curation:** W. Zhou.

**Formal analysis:** L. I. Bendell.

**Funding acquisition:** L. I. Bendell.

**Investigation:** L. I. Bendell, E. LeCadre.

**Methodology:** E. LeCadre.

**Supervision:** L. I. Bendell.

**Writing – original draft:** L. I. Bendell.

**Writing – review & editing:** L. I. Bendell.

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
