## [Decision Letter · Decision Letter 0]

28 Jan 2020

PONE-D-19-33760

Use of sediment dwelling bivalves as biomonitors of plastic particle pollution in intertidal regions; a review and study.

PLOS ONE

Dear Professor Bendell,

Thank you for submitting your manuscript to PLOS ONE. After careful consideration, we feel that it has merit but does not fully meet PLOS ONE’s publication criteria as it currently stands. Therefore, we invite you to submit a revised version of the manuscript that addresses the points raised during the review process.

We would appreciate receiving your revised manuscript by Mar 13 2020 11:59PM. To enhance the reproducibility of your results, we recommend that if applicable you deposit your laboratory protocols in protocols.io, where a protocol can be assigned its own identifier (DOI) such that it can be cited independently in the future. For instructions see: http://journals.plos.org/plosone/s/submission-guidelines#loc-laboratory-protocols

We look forward to receiving your revised manuscript.

Kind regards,

Amitava Mukherjee, ME, Ph.D.

Academic Editor

PLOS ONE

Journal Requirements:

2. In your Methods section, please provide additional location information, including geographic coordinates for the data set if available.

4. We note that Figure 1 in your submission contains map/satellite images which may be copyrighted.

'...Funding from the National Research Council of Canada in the form of a Discovery grant to LB is also acknowledged.

'NO. The funders had no role in study design, data collection and analysis, decision to publish, or preparation of the manuscript.'

Please amend your Financial disclosure statement to declare sources of funding, or state that the authors received no specific funding.

Please provide an amended Funding Statement that declares *all* the funding or sources of support received during this specific study (whether external or internal to your organization) as detailed online in our guide for authors at http://journals.plos.org/plosone/s/submit-nowPlease state what role the funders took in the study.  If any authors received a salary from any of your funders, please state which authors and which funder. If the funders had no role, please state: "The funders had no role in study design, data collection and analysis, decision to publish, or preparation of the manuscript."

6. Please upload a copy of Figures 7 and 8, to which you refer in your tex. If either figure is no longer to be included as part of the submission please remove all reference to it within the text.

7. Please upload a new copy of Figure 1 as the detail is not clear. Please follow the link for more information: http://blogs.PLOS.org/everyone/2011/05/10/how-to-check-your-manuscript-image-quality-in-editorial-manager/

8. Please include a copy of Tables 2-5 in the manuscript, which you refer to in your text, i.e. please add titles to all of the included tables.

9. Please include captions for your Supporting Information files at the end of your manuscript, and update any in-text citations to match accordingly. Please see our Supporting Information guidelines for more information: http://journals.plos.org/plosone/s/supporting-information

Reviewers' comments:

Reviewer's Responses to Questions

**Comments to the Author**

1. Is the manuscript technically sound, and do the data support the conclusions?

Reviewer #1: Yes

2. Has the statistical analysis been performed appropriately and rigorously? 

Reviewer #1: Yes

3. Have the authors made all data underlying the findings in their manuscript fully available?

Reviewer #1: Yes

4. Is the manuscript presented in an intelligible fashion and written in standard English?

Reviewer #1: Yes

5. Review Comments to the Author

Reviewer #1: The present study reviewed the monitored types of MPs and the use of bivalves for biomonitoring of MPs in aquatic ecosystems, and then the authors analyzed the characteristics of MPs from two species of clam sampled in field. The subject of research is interesting. However, this manuscript has some problems with the experimental design. The key point in the present study is to determine whether the sediment dwelling varnish and Manila clam could possibly be good choices for biomonitoring of MP. Therefore the authors determined the characteristics of MPs in the clams. Actually, the information of MPs in the sediment is also very important to support the author’s opinion (…that the proposed organisms should ingest without bias the majority of plastic particles to which they are exposed (P7)). We do not know if the both clams did show bias in MPs particle selection with the MPs information in sediments. Therefore, the authors should provide the information on MPs in the sediments.

Specific comments

1.References: please keep the format consistent of the references.

2.Tables: please redraw all the tables, since the format is not appropriate.

3.Figures: It is suggested that figures 2 and 4 be used as supporting information. Figure 5, please indicate which one is 5a or 5b. Figure 6, the resolution of the third one is not enough.

6. PLOS authors have the option to publish the peer review history of their article (what does this mean?). If published, this will include your full peer review and any attached files.

Reviewer #1: Yes: Jun Bo

---

## [Author Response · Author response to Decision Letter 0]

21 Feb 2020

Response to Editor and Reviewers Comments.

Dear Sir;

Thank you for considering our paper to be included in the PLoS One collection of papers within the theme of “Plastics in the Environment”. 

Within this revised MS we have addressed all edits and comments of the editor and reviewer.

Revisions include;

1) The revised MS has undergone a thorough editing.

2) We have edited the MS to Meet PLOS ONE’s style requirements.

3) Within methods we have included GPS coordinates of the 8 intertidal regions.

4) We have redrafted Figure 1 using maps from the USGS public domain as advised.

5) We have removed funding information from the Acknowledgements

Amended Funding Statement:

Funding from the National Research Council of Canada (NSERC Grant # 31-611307 to LB) is gratefully acknowledge. The funders had no role in study design, data collection and analysis, decision to publish, or preparation of the manuscript.

6) Copies of all figures have been uploaded.

7) Figure 1 has been redrafted for clarity.

8) All Tables are included within the text as with their titles.

9) Captions for SI have been provided at the end of the MS. 

Reviewers Comments.

1) The study objective was to determine if bivalves, not sediments could be used as indicators of MPs within a particular region. However, information on the concentration of MPs within sediments is provided in Kazmiruk et al (2018) for Baynes Sound and this is referenced within the manuscript. Indeed, the information of Kazmiruk et al. (2018) was the reason for choosing the 3 sampling regions within Baynes Sound. Recovery of plastic fragments and spheres from bivalves sampled from these 3 sites confirms the findings of Kazmiruk et al. (2018) and supports the use of bivalves of biomonitors of microplastics within sedimentary environments.

2) References have been reformatted to meet PLOS ONE’s formatting requirements.

3) Tables have all been reformatted.

4) We would like to include Figs 2 and 4 within the main text as these visuals are important information for the reader. The resolution of the figures has been corrected and the titles edited to indicate figure content.

---

## [Decision Letter · Decision Letter 1]

24 Apr 2020

Use of sediment dwelling bivalves to biomonitor of plastic particle pollution in intertidal regions; a review and study.

PONE-D-19-33760R1

Dear Dr. Bendell,

We are pleased to inform you that your manuscript has been judged scientifically suitable for publication and will be formally accepted for publication once it complies with all outstanding technical requirements.

With kind regards,

Amitava Mukherjee, ME, Ph.D.

Academic Editor

PLOS ONE

Additional Editor Comments (optional):

Reviewers' comments:

Reviewer's Responses to Questions

**Comments to the Author**

1. If the authors have adequately addressed your comments raised in a previous round of review and you feel that this manuscript is now acceptable for publication, you may indicate that here to bypass the “Comments to the Author” section, enter your conflict of interest statement in the “Confidential to Editor” section, and submit your "Accept" recommendation.

Reviewer #1: All comments have been addressed

2. Is the manuscript technically sound, and do the data support the conclusions?

Reviewer #1: Yes

3. Has the statistical analysis been performed appropriately and rigorously? 

Reviewer #1: Yes

4. Have the authors made all data underlying the findings in their manuscript fully available?

Reviewer #1: Yes

5. Is the manuscript presented in an intelligible fashion and written in standard English?

Reviewer #1: Yes

6. Review Comments to the Author

Reviewer #1: (No Response)

7. PLOS authors have the option to publish the peer review history of their article (what does this mean?). If published, this will include your full peer review and any attached files.

Reviewer #1: No

---

## [Editor Report · Acceptance letter]

29 Apr 2020

PONE-D-19-33760R1 

Use of sediment dwelling bivalves to biomonitor plastic particle pollution in intertidal regions; a review and study. 

Dear Dr. Bendell:

I am pleased to inform you that your manuscript has been deemed suitable for publication in PLOS ONE. Congratulations! Your manuscript is now with our production department. 

With kind regards,

on behalf of

Professor Dr. Amitava Mukherjee 

Academic Editor

PLOS ONE